# Circulating Cell-Free DNA Levels in Psychiatric Diseases: A Systematic Review and Meta-Analysis

**DOI:** 10.3390/ijms24043402

**Published:** 2023-02-08

**Authors:** Mark M. Melamud, Valentina N. Buneva, Evgeny A. Ermakov

**Affiliations:** 1Institute of Chemical Biology and Fundamental Medicine, Siberian Branch of the Russian Academy of Sciences, 630090 Novosibirsk, Russia; 2Department of Natural Sciences, Novosibirsk State University, 630090 Novosibirsk, Russia

**Keywords:** cfDNA, cf-mtDNA, DAMPs, schizophrenia, bipolar disorder, major depressive disorder, inflammation

## Abstract

The cell-free DNA (cfDNA) levels are known to increase in biological fluids in various pathological conditions. However, the data on circulating cfDNA in severe psychiatric disorders, including schizophrenia, bipolar disorder (BD), and depressive disorders (DDs), is contradictory. This meta-analysis aimed to analyze the concentrations of different cfDNA types in schizophrenia, BD, and DDs compared with healthy donors. The mitochondrial (cf-mtDNA), genomic (cf-gDNA), and total cfDNA concentrations were analyzed separately. The effect size was estimated using the standardized mean difference (SMD). Eight reports for schizophrenia, four for BD, and five for DDs were included in the meta-analysis. However, there were only enough data to analyze the total cfDNA and cf-gDNA in schizophrenia and cf-mtDNA in BD and DDs. It has been shown that the levels of total cfDNA and cf-gDNA in patients with schizophrenia are significantly higher than in healthy donors (SMD values of 0.61 and 0.6, respectively; *p* < 0.00001). Conversely, the levels of cf-mtDNA in BD and DDs do not differ compared with healthy individuals. Nevertheless, further research is needed in the case of BD and DDs due to the small sample sizes in the BD studies and the significant data heterogeneity in the DD studies. Additionally, further studies are needed on cf-mtDNA in schizophrenia or cf-gDNA and total cfDNA in BD and DDs due to insufficient data. In conclusion, this meta-analysis provides the first evidence of increases in total cfDNA and cf-gDNA in schizophrenia but shows no changes in cf-mtDNA in BD and DDs. Increased circulating cfDNA in schizophrenia may be associated with chronic systemic inflammation, as cfDNA has been found to trigger inflammatory responses.

## 1. Introduction

Cell-free DNA (cfDNA) is detectable fragments of nucleic acids released from the cells into the circulation [1]. CfDNA was first identified in the blood of a patient with leukemia by Labbe et al. in 1931 [2], and then Mandel and Métais in 1948 found cfDNA in the plasma of healthy donors [3]. These pioneering studies have shown that endogenous nucleic acids are permanently present in the bloodstream and that their quantities are changed in pathological conditions.

The heterogeneous fraction of circulating cfDNA includes genomic (cf-gDNA) and mitochondrial DNA (cf-mtDNA), depending on the origin [4]. The sizes of cfDNA also vary significantly (80–10,000 bp, with normal peaks of 166 bp fragments for cf-gDNA and 20–100 bp for cf-mtDNA), depending on the mechanisms of fragmentation and release [5]. The release of cfDNA from cells can occur passively as a result of various forms of cell death or via active secretion [1,6]. Most cfDNA enters the bloodstream through the secretion of extracellular vesicles, including exosomes, microparticles, or apoptotic bodies [7]. Cell death caused by apoptosis, necrosis, and pyroptosis contributes to the release of cfDNA into the bloodstream [1,8]. Programmed neutrophil death (NETosis) leading to the extrusion of nuclear-derived decondensed DNA (neutrophil extracellular traps, NETs) is also a source of cfDNA [9]. Additionally, autophagy, erythroblast enucleation, and other processes promote the release of cfDNA into circulation [1]. Thus, the majority of cfDNA in circulation is believed to originate from hematopoietic cells, including leukocytes and erythrocyte progenitors [10]. However, under certain physiological or pathological conditions, a specific cfDNA of a different origin appears, such as from tumor cells in cancer or a damaged organ in trauma.

The interest in using cfDNA analyses to obtain diagnostic information has recently been growing [11]. The analysis of cfDNA in biological fluids is considered a form of “liquid biopsy” because this approach allows for obtaining diagnostic information without invasive technologies [8,11]. The analysis of cfDNA has shown efficacy in prenatal screening for fetal chromosomal aneuploidies [12], cancer diagnosis [13], and the assessment of transplant rejection [14]. Interestingly, in addition to data on the quantitative levels and genetic markers of circulating cfDNA, the differences in DNA methylation patterns, fragmentation profiles, and topological properties can also be used for diagnostic purposes [11]. The analysis of cfDNA may also be useful in mental disorders.

The level of cfDNA is also of interest in association with inflammatory conditions. CfDNA is known to be part of the damage-associated molecular patterns (DAMPs) [15]. CfDNA can be recognized by pattern recognition receptors (PRRs) promoting sterile inflammation [16]. Inflammation is known to be associated with the pathogenesis of severe mental illnesses such as schizophrenia [17], bipolar disorder (BD) [18], and depressive disorders (DDs) [19]. However, the inflammation triggers in psychiatric disorders are still poorly understood, and cfDNA may be one of them.

Thus, cfDNA may be a promising biomarker for mental disorders. However, the data on the circulating cfDNA levels in psychiatric disorders are rare and conflicting. According to our data, there have been no systematic reviews analyzing the level of cfDNA in certain mental disorders. We found only two systematic reviews in which data on the cf-mtDNA levels in schizophrenia, BD, DDs, and other disorders were considered together, not allowing an assessment of the cf-mtDNA levels in individual diseases [20,21]. New studies that were not included in these meta-analyses have recently been published. Additionally, the levels of different types of cfDNA (cf-mtDNA and cf-gDNA) or total cfDNA) in psychiatric disorders have not been analyzed in meta-analyses. Consequently, this systematic review and meta-analysis aimed to analyze the circulating cfDNA (cf-mtDNA, cf-gDNA, or total cfDNA) levels in schizophrenia, BD, and DDs separately for each disease compared with healthy subjects.

## 2. Methods

### 2.1. Study Search

This systematic review and meta-analysis was performed following the Preferred Reporting Items for Systematic Reviews and Meta-Analyses (PRISMA 2020) guidelines [22,23]. The search for eligible articles was carried out in the PubMed/MEDLINE, Google Scholar, Scopus, and Web of Science databases, as well as using query tools in the Google search engine. The time frame for the literature search was from January 1948 (cfDNA discovery date) to June 2022. The following combinations of keywords with Boolean operators (OR, AND) were used in the search queries (cell-free DNA OR CfDNA OR Cell-free mitochondrial DNA OR Cf-mtDNA OR CfmtDNA OR DAMP) AND (schizophrenia OR bipolar disorder OR BD OR major depressive disorder OR MDD OR depression OR psychosis OR psychiatric diseases). A reference list of selected articles was screened to identify additional studies. Automation tools were not used to exclude irrelevant records. Only published articles were considered. Conference abstracts and editorials were not included in the study. Two investigators (M.M.M. and E.A.E.) independently reviewed the results of each query and then assessed the relevance of all retrieved articles. The final reference list was compiled after reaching consensus with all co-authors.

### 2.2. Eligibility Criteria

The following inclusion criteria were used in this meta-analysis: (i) the study (or part of it) should be devoted to the measurement of the cfDNA concentration in the serum or plasma of patients with verified diagnoses of schizophrenia, BD, or DDs, as well as healthy donors; (ii) the study should be conducted on groups of people 18 years of age and older; (iii) the study must be indexed in at least one of the international databases. The meta-analysis included studies published in English or other languages with English translations. No further subdivision into groups (e.g., by disease form) was performed due to the small number of studies that met the inclusion criteria. The study did not include the results for patients after therapy; the groups marked as “after therapy” in the reports retrieved were excluded from the analysis. Data from the total patient population were included in the meta-analysis. Data from subgroups of patients were not included. Different types of cfDNA (cf-mtDNA, cf-gDNA) or total cfDNA were analyzed separately.

### 2.3. Data Extraction and Statistical Analysis

Two independent reviewers (M.M.M. and E.A.E.) extracted data from the retrieved articles. Some studies [24,25] did not have all of the necessary data for the statistical analysis (the arithmetic mean and standard deviation were missing). The required data for these articles were predicted based on the median and interquartile range, according to the methodology described by Weir et al. [26]. The effect size was estimated using the standardized mean difference (SMD), since the cfDNA concentration was measured with different units of measurement (ng/mL, nM, copies/µL, GE/mL) in the analyzed reports.

The results were analyzed using Review Manager 5.4.1, MedCalc, and OriginPro 2021 software. A random-effects model was used to pool the effect sizes. The degree of heterogeneity was assessed using the I2 statistics test [27]. I^2^ values of 0–25%, 26–50%, 51–75%, and 75–100% were classified as indicating no, low, moderate, and substantial heterogeneity, respectively [27]. The level of bias was assessed using Egger’s test [28], Begg’s test [29], and funnel plots.

## 3. Results

### 3.1. Reports Selection

The PRISMA flowchart diagram created using the web-based Shiny app is presented in Figure 1 [30]. In the primary search, 2997 reports were selected based on the title (1090, 487, and 1420 for schizophrenia, BD, and DDs, respectively). Irrelevant reports were excluded after reviewing the title and abstracts. In sum, 11 papers were retrieved for schizophrenia, 4 for BD, and 7 for DDs. When re-examining the texts of the retrieved papers, one study was excluded from the schizophrenia group describing an increase in cfDNA concentration in children [31], along with one study from the DDs group describing depression in older people (late-life depression) [32]. These studies were excluded because young or old age may affect cfDNA levels [33]. Two reports on cf-mtDNA in schizophrenia were relevant [34,35]. However, they were also excluded because two reports were not enough to conduct a meta-analysis for this type of cfDNA (see further). For the same reason, one report on cf-gDNA was excluded from the DDs group [36], as the remaining studies analyzed cf-mtDNA. Ultimately, the following numbers of reports were included in the meta-analysis: schizophrenia, 8 [24,36,37,38,39,40,41,42]; BD, 4 [25,35,43,44]; DDs, 5 [35,45,46,47,48]. The total number of individual studies included in the meta-analysis was sixteen because some reports considered several diseases simultaneously.

### 3.2. Reports Characteristics

The list of reports included in the meta-analysis and their characteristics are summarized in Table 1. Various analytical methods allow the detection of different types of cfDNA (cf-mtDNA and cf-gDNA) or total cfDNA. Therefore, we divided all reports depending on the type of analyzed cfDNA. As a result, the total cfDNA was investigated in 6 studies [37,38,39,40,41,42] and cf-gDNA in three reports in the case of schizophrenia [24,36,42]. There were insufficient data for a meta-analysis of cf-mtDNA in schizophrenia (only two studies) [34,35]. Cf-mtDNA was analyzed in BD and DDs. The data for other types of cfDNA were insufficient for a meta-analysis in BD and DDs. In particular, only one study on cf-gDNA was found for DDs [36]. Accordingly, this meta-analysis analyzed the total cfDNA and cf-gDNA in schizophrenia, and cf-mtDNA was investigated in BD and DDs.

There was some heterogeneity across disease groups (Table 1). Among the eight reports in the schizophrenia group, one report examined total the cfDNA and cf-gDNA in first-episode psychosis (FEP) patients [42]. In the DDs group, three reports included patients diagnosed with major depressive disorder (MDD) [35,47,48], one report included individuals with current depression [46], and one study included suicide attempters [45]. We accounted for this heterogeneity in the meta-analysis (see below).

Plasma, rather than serum, is known to be the preferred source for cfDNA analyses [49]. The blood plasma was analyzed in most of the reports included in the meta-analysis (Table 1). In particular, only 2 out of 8 reports in schizophrenia [24,36], 2 out of 4 in BD [25,43], and only 1 out of 5 in DDs [48] were devoted to the analysis of cfDNA in serum. Plasma was used in all reports of total cfDNA in schizophrenia. Serum was used in 2 out of 3 reports of cf-gDNA in schizophrenia.

The cfDNA isolation and detection methods also differed in the included reports (Table 1). The column method for cfDNA isolation was used in all reports for BD and DDs. In the case of schizophrenia, three of the eight studies used the column method [24,36,38], four used organic solvents [37,39,40,41], and one used magnetic particles [42] for DNA extraction. The fluorescent detection method was primarily used to analyze the total cfDNA in schizophrenia (only one report used fluorescence correlation spectroscopy) [38]. The quantitative real-time polymerase chain reaction (qPCR) method was mainly used to determine cf-gDNA in schizophrenia and cf-mtDNA in BD and DDs. Alu repeat amplification was used for the cf-gDNA analysis with qPCR [24,36]. The target genes for the cf-mtDNA analysis were as follows: *MT-ATP8*, *MT-ND1*, *MT-ND2*, and *MT-ND4*. The cfDNA concentration was presented in various units of measurement. Eight studies reported cfDNA concentrations as ng/mL (or pg/mL) [24,36,37,39,40,41,42,48], five as copies/µL [35,43,44,46,47], two as GE/mL (or U/mL) [25,45], and one as nM [38]. All selected studies were published between 2015 and 2022.

### 3.3. CfDNA Level in Schizophrenia

As stated above, the total cfDNA and cf-gDNA were analyzed in schizophrenia due to insufficient data existing for another type of cfDNA (cf-mtDNA). A meta-analysis of the circulating total cfDNA in schizophrenia pooled data from six studies with a total of 686 patients and 375 healthy controls. It has been shown that the circulating total cfDNA concentration in schizophrenia is significantly higher than in healthy donors (Figure 2a). The SMD for the overall effect was 0.61 (95% CI = [0.40 to 0.82]), with moderate heterogeneity (Chi^2^ = 11.6, df = 5, *p* < 0.04; I^2^ = 57%). The test for the overall effect also confirmed the significance of the differences (Z = 5.6, *p* < 0.00001). No evidence of publication bias was observed using Egger’s test (*p* = 0.773) and Begg’s test (*p* = 0.851). The funnel plot analysis showed signs of asymmetry (Figure 3a). The three reports on the left side of the graph were from the same research group using a similar method [37,39,41]. Therefore, the observed asymmetry may indicate a publication bias. One report falling outside the confidence interval was probably related to the measurement methodology (fluorescence correlation spectroscopy) [38]. Additionally, there was sample heterogeneity (one of the six reports analyzed FEP patients) [42]. To reduce sample heterogeneity, we excluded this study, but the results were almost unchanged (SMD = 0.62; 95% CI = [0.38 to 0.86] with moderate heterogeneity (Chi^2^ = 11.41, df = 4, *p* < 0.02; I^2^ = 65%); test for overall effect: Z = 5.03, *p* < 0.00001).

The meta-analysis of circulating cf-gDNA in schizophrenia included three studies with a total of 367 patients and 231 healthy individuals. A meta-analysis showed that the cf-gDNA concentration in schizophrenia was significantly higher than in the controls (Figure 2b). The SMD for the overall effect was 0.6 (95% CI = [0.43 to 0.77]). The test for the overall effect also confirmed the significance of the differences (Z = 6.93, *p* < 0.00001). Interestingly, there was practically no heterogeneity in the report results (Chi^2^ = 0.15, df = 2, *p* < 0.93; I^2^ = 0%). However, while Egger’s test (*p* = 0.058) and Begg’s test (*p* = 0.117) showed no evidence of bias, there were some indications of publication bias. In particular, the two reports had the same mean and standard deviation for the cf-gDNA for the group of healthy donors [24,36]. The funnel plot also confirmed this observation (Figure 3b). Therefore, these results must be interpreted with caution. Nevertheless, after removing one of the studies with the same group of healthy donors [24], the meta-analysis results remained significant (SMD = 0.61, 95% CI = [0.38 to 0.84] with no heterogeneity (Chi^2^ = 0.13, df = 1, *p* < 0.72; I^2^ = 0%); test for overall effect: Z = 5.23, *p* < 0.00001). Therefore, further studies are needed to confirm the increased circulating cf-gDNA concentrations in schizophrenia.

### 3.4. CfDNA Level in Bipolar Disorder

Less reliable data were obtained for BD because only four studies were included in the meta-analysis, with a total of 123 patients and 99 healthy controls. Only the cf-mtDNA concentration was analyzed in a meta-analysis. The results indicated that the circulating cf-mtDNA concentration in BD is not statistically different from in healthy donors (Figure 2c). The SMD for the overall effect was −0.06 (95% CI = [−0.55 to 0.42]), with moderate heterogeneity (Chi^2^ = 8.39, df = 3, *p* < 0.04; I^2^ = 64%). The test for the overall effect also showed no significant differences (Z = 0.06, *p* = 0.95). No evidence of publication bias was observed using Egger’s test (*p* = 0.646) and Begg’s test (*p* = 0.467). However, one report in the funnel plot was outside the confidence interval [35], which may indicate a publication bias (Figure 3c). Thus, more research is needed on cf-mtDNA and other types of cfDNA in BD.

### 3.5. CfDNA Level in Depressive Disorders

The meta-analysis in DDs included only data for the circulating cf-mtDNA. A meta-analysis pooled data from five studies with a total of 456 patients and 190 healthy controls. It has been shown that the cf-mtDNA concentrations in DDs are not significantly different from those in healthy donors (Figure 2d). The SMD for the overall effect was −0.03 (95% CI = [−1.04; 0.98]). It is important to note that substantial heterogeneity was observed (Chi^2^ = 107.78, df = 4, *p* < 0.00001; I^2^ = 96%). Egger’s test (*p* = 0.622) and Begg’s test (*p* = 0.2) showed no signs of bias. However, the funnel plot showed signs of publication bias, as many of the studies were outside the confidence intervals (Figure 3d). We further excluded two reports with non-specific depressive symptoms [45,46] and analyzed only reports with MDD to reduce the sample heterogeneity. However, the results have not changed (SMD = −0.16, 95% CI = [−0.43 to 0.11]; test for overall effect: Z = 1.18, *p* = 0.24). The study heterogeneity was also high (Chi^2^ = 67.44, df = 2, *p* < 0.00001; I^2^ = 97%). Thus, further studies are needed to reduce the heterogeneity.

## 4. Discussion

### 4.1. Results Overview

The results of the meta-analysis indicate that the levels of circulating total cfDNA and cf-gDNA in patients with schizophrenia are significantly higher than in healthy donors, but the levels of cf-mtDNA in BD and DDs do not differ in comparison with healthy individuals (Figure 2). These data revealed gaps in knowledge regarding the different types of cfDNA in psychiatric disorders. In particular, the data on cf-mtDNA in schizophrenia are scarce (there are only two conflicting reports) [34,35]. The data on the total cfDNA and cf-gDNA are practically absent in BD and DDs (only one report is devoted to analyzing cf-gDNA in MDD) [36]. Additionally, the data on cf-mtDNA in BD and DDs are heterogeneous and at risk of bias. Therefore, the new research should focus on analyzing cf-mtDNA, cf-gDNA, or the total cfDNA in psychiatric disorders.

The increase in circulating total cfDNA and cf-gDNA in schizophrenia may be associated with the pathogenic mechanisms of the disease. However, the inconsistency in the published reports on BD and DDs may be related to the dynamic nature of the cfDNA release and excretion. The increase in cfDNA may be physiological. For example, cfDNA increases with age and after exercise or acute stress [21]. However, with chronic exposure, the cfDNA concentration can paradoxically decrease due to a compensatory increase in blood nuclease activity and anti-DNA antibody levels [50]. However, high levels of cfDNA are thought to be associated with enhanced cell death or with clearance inefficiency of the circulating cfDNA [8,51]. It can be assumed that these processes occur in schizophrenia. Various processes, including oxidative stress, promote enhanced cell death through apoptosis [1]. Interestingly, oxidative stress and mitochondrial dysfunction are known to be associated with schizophrenia [52]. Signs of increased susceptibility to cell apoptosis (including fibroblasts and cortical neurons) have also been found in schizophrenia [53,54]. Therefore, apoptosis dysfunction may be one reason for the increase in cfDNA in schizophrenia. Additionally, enhanced neutrophil death (NETosis), accompanied by the extrusion of intracellular DNA and the formation of NETs, also increases cfDNA [9]. Three meta-analyses demonstrate increases in neutrophil levels and the neutrophil-to-lymphocyte ratio in the blood of patients with schizophrenia compared with healthy donors [55,56,57]. Therefore, it can be assumed that neutrophils can be one of the sources of circulating cfDNA in schizophrenia. However, hematopoietic and neuronal cells can also be substantial sources of cfDNA [10,42]. Thus, identifying the cellular sources of cfDNA in the blood of patients with schizophrenia is an important avenue for further research.

Reduced nucleic acid clearance may also contribute to increased circulating cfDNA levels [51,58]. Normally, the circulating cfDNA is rapidly cleaved by endonucleases and excreted through the liver, spleen, and kidneys [51]. DNase I and DNase I-like III (DNASE1L3) play a vital role in the degradation of circulating nucleic acids [59]. Nuclease function studies in schizophrenia are rare. There is limited evidence of a high level of plasma endonuclease activity in patients with schizophrenia [37]. Increased nuclease activity may reflect homeostatic compensatory processes in response to increased cfDNA. Further research may help elucidate the functionality of endonucleases in schizophrenia and other psychiatric disorders. There are also data showing that DNASE1L3-deficient mice showed elevated levels of circulating cfDNA and increased production of autoantibodies to DNA and chromatin [58]. Interestingly, a systematic analysis showed a higher prevalence of antinuclear and anti-DNA antibodies in patients with schizophrenia [60]. Thus, the elevated level of cfDNA in schizophrenia may explain the high prevalence of anti-DNA antibodies.

The released cfDNA may be bound with histones, high-mobility group protein B1 (HMGB1), cathelicidin (LL-37), and other proteins [59]. Interestingly, the serum level of HMGB1 in patients with schizophrenia is significantly higher than in healthy donors [61,62]. These data indirectly confirm the high level of cfDNA in schizophrenia. However, cfDNA bound with DNA-binding proteins has great inflammatory potential.

### 4.2. Association with Chronic Low-Grade Inflammation

The recognition of cfDNA through PRRs is known to trigger an inflammatory response [16,63,64]. The chronic proinflammatory state is well documented in schizophrenia [17], BD [18], and MDD [19]. Therefore, based on the data from this meta-analysis, it can be assumed that one of the triggers of chronic low-grade inflammation in schizophrenia is an increased level of cfDNA. CpG-enriched ribosomal cfDNA accumulates in chronic pathologies (including schizophrenia) because it is nuclease-resistant [37]. Interestingly, CpG-enriched cfDNA are highly bioactive [65,66]. Additionally, cfDNA can be oxidized under the chronic oxidative stress conditions observed in schizophrenia [52]. Oxidized cfDNA exhibits proinflammatory properties [67,68]. There is evidence that the cfDNA in patients with schizophrenia stimulated the expression of the *TLR9* and *STING* genes while simultaneously blocking the expression of *RIG-I* and *AIM2*, which are involved in cfDNA sensing and activate the inflammatory response [69]. These data suggest that oxidized/CpG-enriched cfDNA can boost the inflammatory response even in decreased or normal amounts. Thus, neutralizing the proinflammatory effects of cfDNA, such as by using DNase I or nucleic acid-binding polymers [70], may be a promising therapeutic strategy in schizophrenia.

### 4.3. Limitations

The limitations of this meta-analysis are related to the small number of cfDNA studies in psychiatric disorders. The results obtained in this meta-analysis for DDs and BD should be tested in further studies, since the conclusions are based on a small number of reports and a relatively small sample. Some studies were performed by the same research group (in particular, two reports by a group from China [24,36], four reports by a group from Russia [37,39,40,41], two reports by a group from Sweden [46,47], and two reports by a group from Japan [35,44]), so these studies may have had overlapping samples and a risk of publication bias. Additionally, the heterogeneity of the methods used to isolate and determine the concentration of cfDNA in the reports included in the meta-analysis (Table 1) is a limitation. Additionally, we were unable to conduct a subgroup analysis due to the low number of reports. Bias tests have low power with a small number (*n* < 10) of included reports [28,29]. Therefore, the publication bias cannot be excluded entirely. However, the authors strictly followed the PRISMA 2020 guidelines to minimize bias [22,23].

## 5. Conclusions

In summary, this meta-analysis provided evidence for high levels of total cfDNA and cf-gDNA in the plasma and serum of patients with schizophrenia compared with healthy individuals. In contrast, evidence for altered cf-mtDNA levels in BD and DDs was not found in this meta-analysis. However, the lack of significant differences may be partly caused by the small number of studies in BD and the hypothetical publication bias in DDs. Data on other types of cfDNA or total cfDNA in these psychiatric disorders are scarce. In particular, there are few studies on cf-mtDNA in schizophrenia or total cfDNA and cf-gDNA in BD and DDs. Therefore, further research should fill this knowledge gap. High levels of cfDNA in schizophrenia may be associated with chronic systemic inflammation in schizophrenia, as cfDNA has been found to trigger inflammatory responses. The obtained results have opened up many lines of future work. First, it is necessary to uncover the cellular source and release mechanisms of cfDNA in psychiatric disorders. Secondly, the specific features of cfDNA (methylation profile, fragmentation, topology (linear, circular, etc.), and association with proteins) in mental disorders should be investigated. Third, the functionality of the DNases and cfDNA removal mechanisms also needs to be evaluated. Finally, the therapeutic potential of circulating cfDNA-neutralizing agents in schizophrenia to reduce inflammation should be assessed.

## Figures and Tables

**Figure 1 ijms-24-03402-f001:**
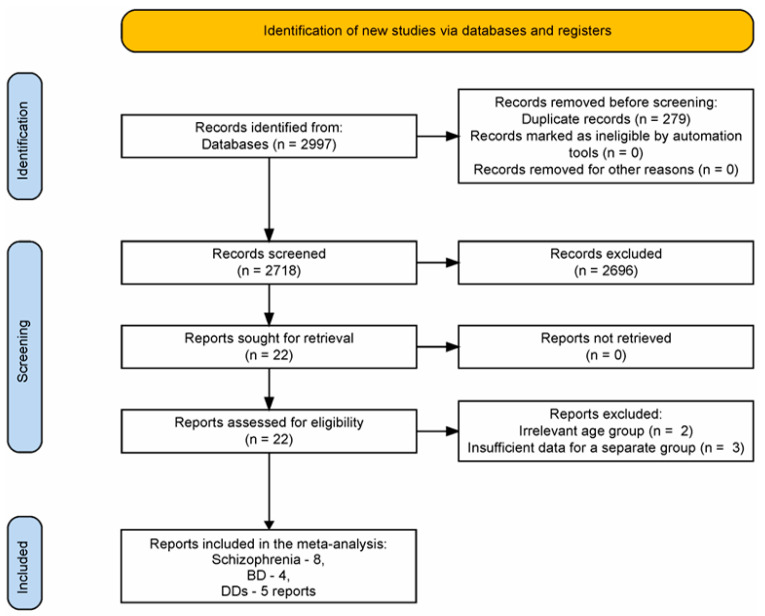
PRISMA flowchart diagram of report screening and selection process.

**Figure 2 ijms-24-03402-f002:**
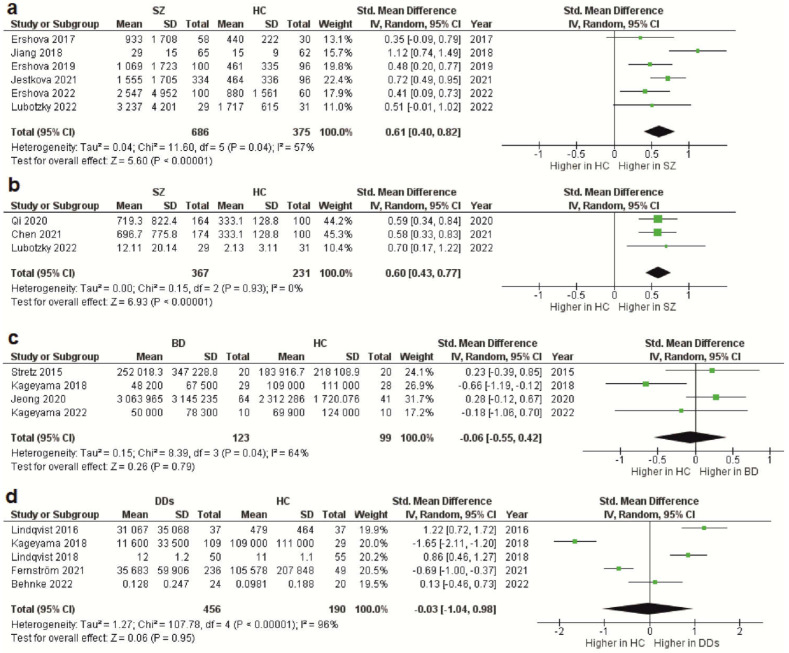
Forest plot showing plasma and serum levels of total cfDNA in patients with schizophrenia (**a**), cf-gDNA in schizophrenia (**b**), cf-mtDNA in BD (**c**), and cf-mtDNA in DDs (**d**) compared with healthy controls. BD—bipolar disorder; CI—confidence interval; DDs—depressive disorders; SD—standard deviation; SZ—schizophrenia; HC—healthy controls.

**Figure 3 ijms-24-03402-f003:**
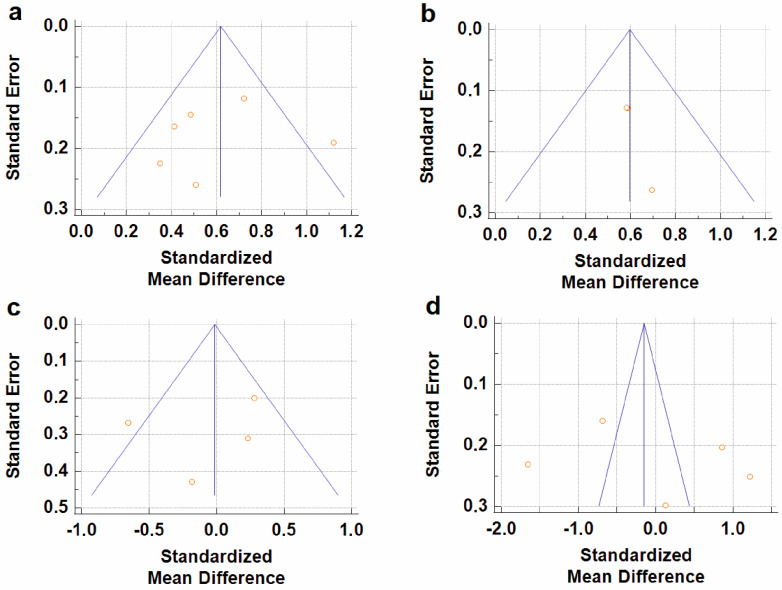
Funnel plots for the analysis of the level of bias in meta-analytical data for total cfDNA in schizophrenia (**a**), cf-gDNA in schizophrenia (**b**), cf-mtDNA in BD (**c**), and cf-mtDNA in DDs (**d**).

**Table 1 ijms-24-03402-t001:** Reports of circulating cfDNA concentrations in schizophrenia, bipolar disorder, and depressive disorders included in the meta-analysis.

Study	Year	DNA Type	Sample	Extraction Method	Detection Method	Population
Schizophrenia						
Ershova et al. [39]	2017	Total CfDNA	Plasma	Solvent extraction method	FL, PicoGreen dye	SZ 58/HC 30
Jiang et al. [38]	2018	Total CfDNA	Plasma	TIANamp Micro DNA Kit (spin-column)	FCS	SZ 65/HC 62
Ershova et al. [37]	2019	Total CfDNA	Plasma	Solvent extraction method	FL, PicoGreen dye	SZ 100/HC 96
Jestkova et al. [40]	2021	Total CfDNA	Plasma	Solvent extraction method	FL, PicoGreen dye	SZ 334/HC 95
Ershova et al. [41]	2022	Total CfDNA	Plasma	Solvent extraction method	FL, PicoGreen dye	SZ 100/HC 60
Lubotzky et al. [42]	2022	Total CfDNA and Cf-gDNA	Plasma	QIAsymphony DSP Circulating DNA Kit (magnetic particles)	FL, bisulfite DNA treatment, PCR amplification followed by NGS	FEP 29/HC 31
Chen et al. [24]	2021	Cf-gDNA	Serum	TianLong DNA Kit (spin-column)	qPCR, target: Alu repeats	SZ 174/HC 100
Qi et al. [36]	2020	Cf-gDNA	Serum	TianLong DNA Kit (spin-column)	qPCR, target: Alu repeats	SZ 164/HC 100
Bipolar Disorder						
Stertz et al. [25]	2015	Cf-mtDNA	Serum	QIAmp DNA Mini Kit (spin-column)	qPCR, target: *MT-ATP8* gene	BD 20/HC 20
Kageyama et al. [35]	2018	Cf-mtDNA	Plasma	QIAamp DNA Blood Mini Kit (spin-column)	qPCR, target: *MT-ND1* and *MT-ND4* genes	BD 28/HC 29
Jeong et al. [43]	2020	Cf-mtDNA	Serum	QIAmp DNA Mini Kit (spin-column)	qPCR, target: *MT-ND1* gene	BD 64/HC 41
Kageyama et al. [44]	2022	Cf-mtDNA	Plasma	QIAamp DNA Blood Mini Kit (spin-column)	qPCR, target: *MT-ND1* and *MT-ND4* genes	BD 10/HC 10
Depressive disorders						
Lindqvist et al. [45]	2016	Cf-mtDNA	Plasma	QIAmp 96 DNA Blood Kit (spin-column)	qPCR, target: *MT-ND2* gene	Suicide attempters 37/HC 37
Kageyama et al. [35]	2018	Cf-mtDNA	Plasma	QIAamp DNA Blood Mini Kit (spin-column)	qPCR, target: *MT-ND1* and *MT-ND4* genes	MDD 109/HC 29
Lindqvist et al. [47]	2018	Cf-mtDNA	Plasma	QIAmp 96 DNA Blood Kit (spin-column)	qPCR, target: *MT-ND1* and *MT-ND4* genes	MDD 50/HC 55
Fernström et al. [46]	2021	Cf-mtDNA	Plasma	QIAmp DNA Blood Mini Kit (spin-column)	qPCR, target: *MT-ND2* gene	Current depression 236/HC 49
Behnke et al. [48]	2022	Cf-mtDNA	Serum	QIAamp DNA Micro Kit (spin-column)	qPCR with multiple target	MDD 24/HC 20

Abbreviations: BD—bipolar disorder; CfDNA—cell-free DNA; Cf-mtDNA—mitochondrial cell-free DNA; Cf-gDNA—genomic cell-free DNA; FEP—first-episode psychosis; FL—fluorescent detection method; FCS—fluorescence correlation spectroscopy; HC—healthy controls; MDD—major depressive disorder; *MT-ATP8*—mitochondrially encoded ATP synthase membrane subunit 8 gene; *MT-ND1*, *MT-ND2*, *MT-ND4* genes—mitochondrially encoded NADH–ubiquinone oxidoreductase core subunit 1 (or 2 or 4, respectively) genes; NGS—next-generation sequencing; SZ—schizophrenia; qPCR—quantitative real-time polymerase chain reaction.

## Data Availability

All data analyzed during this study are publicly available and have been previously published. References to these works are included in this article. Additional information is available on request from the corresponding author.

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
