# Peer review of "Circulating Cell-Free DNA Levels in Psychiatric Diseases: A Systematic Review and Meta-Analysis"

_ijms, 2023, doi:10.3390/ijms24043402_

Round 1
Reviewer 1 Report
The manuscript by Melamud et al. entitled “Circulating cell-free DNA level in psychiatric diseases: a 2 systematic review and meta-analysis” represents the meta-analysis clarifying the differences in the level of cfDNA (cf-gDNA, cf-mtDNA) between patients of several psychiatric disorders and healthy donors based on existing studies in this field. Since the published data in this field of research related to psychopathologies of interest is scarce, the authors’ findings are preliminary; however, they are interesting. This is honestly a very well-founded, well-written and thorough study.
However, there are some minor points that have to be clarified and corrected:
1. In the Abstract the authors indicated that “In conclusion, this meta-analysis provided the first evidence of an increase in total cfDNA and cf-gDNA in schizophrenia but found no change of cf-mtDNA in BD and DDs.” Based on this sentence, the question arises what was revealed for all mentioned types of cfDNA in all three examined psychiatric diseases? I would like to suggest to include some information on the groups analyzed within the present study (i.e. total cfDNA and cf-gDNA in schizophrenia, and cf-mtDNA was investigated in BD and DDs) in Abstract.
2. The total number of included studies in meta-analysis differs between Fig.1 and in subsection 3.1, line 144. Please clarify.
3. I would suggest to add in Conclusion the possible explanation of absent evidence for altered cf-mtDNA levels in DDs, which can be partially caused by publication bias among the selected studies, i.e. only one article coincided confidence intervals in the funnel plot.
4. The authors reported that they have accounted for heterogeneity in the groups comprising DDs in the meta-analysis. However, please clarify, if you have also accounted for heterogeneity attributed to the differences of cfDNA measurement methods and the concentration units in meta-analysis?
Overall, this is a good study. Therefore, I suggest accept after minor revision, providing that the authors will address these comments above.
Author Response
Dear Reviewer,
The authors deeply appreciate your thorough analysis of manuscript.
Below we answer your suggestions point by point. Your comments are in italics.
The manuscript by Melamud et al. entitled “Circulating cell-free DNA level in psychiatric diseases: a 2 systematic review and meta-analysis” represents the meta-analysis clarifying the differences in the level of cfDNA (cf-gDNA, cf-mtDNA) between patients of several psychiatric disorders and healthy donors based on existing studies in this field. Since the published data in this field of research related to psychopathologies of interest is scarce, the authors’ findings are preliminary; however, they are interesting. This is honestly a very well-founded, well-written and thorough study.
Reply: Thank you for your high appreciation of our manuscript.
However, there are some minor points that have to be clarified and corrected:
- In the Abstract the authors indicated that “In conclusion, this meta-analysis provided the first evidence of an increase in total cfDNA and cf-gDNA in schizophrenia but found no change of cf-mtDNA in BD and DDs.” Based on this sentence, the question arises what was revealed for all mentioned types of cfDNA in all three examined psychiatric diseases? I would like to suggest to include some information on the groups analyzed within the present study (i.e. total cfDNA and cf-gDNA in schizophrenia, and cf-mtDNA was investigated in BD and DDs) in Abstract.
Reply: Thank you for your suggestion. Indeed, the Abstract was not written clearly enough. We have added the following sentence to Abstract: “However, there was only enough data to analyze total cfDNA and cf-gDNA in schizophrenia and cf-mtDNA in BD and DDs”. We also moved one sentence closer to the end of the Abstract: “Data on the levels of other types of cfDNA in these diseases were insufficient for analysis. In our opinion, the Abstract has now become more understandable.
- The total number of included studies in meta-analysis differs between Fig.1 and in subsection 3.1, line 144. Please clarify.
Reply: The total number of individual studies included in the meta-analysis is sixteen because some reports considered several diseases simultaneously. In particular, the level of cf-mtDNA in BD and DDs was analyzed by Kageyama et al. Therefore, the total number of studies (8+4+5=17) differs from the number of individual studies included in the meta-analysis. In order not to mislead the reader, we corrected Figure 1 and removed the total number of studies. The total number of studies included in the meta-analysis is indicated in the text.
- I would suggest to add in Conclusion the possible explanation of absent evidence for altered cf-mtDNA levels in DDs, which can be partially caused by publication bias among the selected studies, i.e. only one article coincided confidence intervals in the funnel plot.
Reply: This information is detailed in section 3.5. But we have added the following sentence to Conclusion: “However, the lack of significant differences may be partly caused by small number of studies in BD and hypothetical publication bias in DDs”.
- The authors reported that they have accounted for heterogeneity in the groups comprising DDs in the meta-analysis. However, please clarify, if you have also accounted for heterogeneity attributed to the differences of cfDNA measurement methods and the concentration units in meta-analysis?
Reply: Of course, we accounted for heterogeneity associated with the differences in cfDNA measurement methods and the concentration units. We used the standardized mean difference (SMD) calculation method (please see lines 119-121). For this, mean differences must be divided by their respective standard deviations (SD). This will ensure that all values are normalized and will not depend on units of measurement. SMDs can be pooled in meta-analysis because the unit is uniform across studies. The use of SMD is a routine approach for many meta-analyses. If you are interested in the SMD methodology, we can recommend the following review article about this method: “Andrade, C. (2020). Mean difference, standardized mean difference (SMD), and their use in meta-analysis: as simple as it gets. The Journal of clinical psychiatry, 81(5), 11349”.
Overall, this is a good study. Therefore, I suggest accept after minor revision, providing that the authors will address these comments above.
Reply: Thanks again for your valuable suggestions.
Best regards
Authors
Reviewer 2 Report
In their manuscript, M.Melamud and colleagues presented a meta-analysis of available data on circulating cfDNA levels in some major psychiatric disorders, such as schizophrenia, bipolar disorder and depression. Conducted in accordance with the PRISMA 2020 guidelines, the study yielded a useful overview to reveal the current knowledge gaps. I consider the research valuable and worth accepting, especially after implementing some minor suggestions below that I believe to amend the manuscript essentially. The literature references are given simply as PMID ## for the sake of space economy (each reference mentioned below should be considered just as an information source for the authors’ attention, not suggested for citation, though authors may voluntary deem useful to cite some).
1. The importance of studying cfDNA content in psychiatric disorders would be proven and underlined in the context of plasmapheresis as treatment modality for schizophrenia (for instance, PMID: 32404224), although this is still to find the deleterious fraction of circulating molecules involved in the therapeutic effect: cytokines, antibodies, or nucleic acids. To some extent, the answer can be found in a recent study PMID: 35328103.
2. The authors restricted the analysis with the contents of three different kinds of cfDNA. Each kind is considered as a uniform substance, which can be either normal or elevated and exerts always the same dose-depending effect. This approach seems mechanistic and excessively simplifying. First of all, the authors may want to mention more deeply the dynamic nature of cfDNA formation and excretion (they briefly mentioned cfDNA removal mechanisms in line 346). There are some reports that cfDNA can be increased in acute and paradoxically decreased in chronic impacts due to the nuclease and antibody action (PMID: 26113293). This can explain the inconsistency between published reports. Secondly, pathologies and long accumulation processes change the cfDNA composition and properties. As GC-rich regions are more resistant to decomposition and nuclease activity, their fraction accrues, especially rDNA (PMID: 27648955, 31467866). Interestingly, the GC-rich fraction has been shown to be especially bioactive (PMID: 31205871, 26273425, 27753029, 22594608). Another strongly activating modification is cfDNA oxidation (PMID: 32677958, 23644378, 27753029), that can be often found in pathology due to intrinsic oxidative stress. It suggests that oxidized/CG-enriched cfDNA can boost the inflammatory and other responses even in decreased or normal amounts. This is a serious limitation of the study.
3. Another limitation is considering multiple reports published by the same research team as independent reports. There are two studies by Kageyama et al., three by Ershova et al. and two reports by Lindqvist et al. The authors only found ‘asymmetry’ in three reports made by the same research group (Ershova et al.), but attributed it to using ‘a similar method’ (lines 205-207). The authors did not explain in the text what the ‘asymmetry’ means, but isn’t it obvious that using the same technique in not the only cause of the bias? Another factor can be recruitment of the same subjects, for example, in a series of studies, more recent studies performed on a large sample can include data obtained earlier on a smaller sample. So, some measurements in the series are repeated, not independent.
5. Finally, some minor suggestions to repair small defects:
5.1. I found some grammar mistakes. The text should be thoroughly edited by a native English speaker or a professional translator.
5.2. In lines 144-145, the authors declare that ‘The total number of individual studies included in the meta-analysis is sixteen because some reports considered several diseases simultaneously.’ But in Fig. 1 (PRISMA flowchart diagram), the last step box reads ‘Reports included in the meta-analysis (n=17)’. Was the report number 16 or 17? The discrepancy must be corrected.
5.3. In ‘Depressive disorders’ part of Table 1, the study ‘Lindqvist et al. [45]’ is mentioned twice. Apparently, the study by Lindqvist et al. dated 2018 must be reference [47], not [45].
Author Response
Dear Reviewer,
We thank the reviewer for the positive evaluation of our study and helpful criticisms and valuable suggestions. We significantly modified our manuscript. We believe that these changes have significantly improved our manuscript.
Below we answer your suggestions point by point. Your comments are in italics.
In their manuscript, M.Melamud and colleagues presented a meta-analysis of available data on circulating cfDNA levels in some major psychiatric disorders, such as schizophrenia, bipolar disorder and depression. Conducted in accordance with the PRISMA 2020 guidelines, the study yielded a useful overview to reveal the current knowledge gaps. I consider the research valuable and worth accepting, especially after implementing some minor suggestions below that I believe to amend the manuscript essentially. The literature references are given simply as PMID ## for the sake of space economy (each reference mentioned below should be considered just as an information source for the authors’ attention, not suggested for citation, though authors may voluntary deem useful to cite some).
Reply: Thank you for your thorough analysis and high appreciation of the manuscript.
- The importance of studying cfDNA content in psychiatric disorders would be proven and underlined in the context of plasmapheresis as treatment modality for schizophrenia (for instance, PMID: 32404224), although this is still to find the deleterious fraction of circulating molecules involved in the therapeutic effect: cytokines, antibodies, or nucleic acids. To some extent, the answer can be found in a recent study PMID: 35328103.
Reply: We agree that circulating molecules, including cfDNA, may have various physiological effects. Besides, circulating cfDNA can provoke inflammatory responses. We describe it in our manuscript (please see lines 319-322) and refer to work Ershova E. S. et al. (PMID: 35328103). However, plasmapheresis as a treatment for schizophrenia is beyond the scope of our review. In addition, no positive effects of plasmapheresis in schizophrenia were found in the mentioned article (PMID: 32404224).
- The authors restricted the analysis with the contents of three different kinds of cfDNA. Each kind is considered as a uniform substance, which can be either normal or elevated and exerts always the same dose-depending effect. This approach seems mechanistic and excessively simplifying. First of all, the authors may want to mention more deeply the dynamic nature of cfDNA formation and excretion (they briefly mentioned cfDNA removal mechanisms in line 346). There are some reports that cfDNA can be increased in acute and paradoxically decreased in chronic impacts due to the nuclease and antibody action (PMID: 26113293). This can explain the inconsistency between published reports. Secondly, pathologies and long accumulation processes change the cfDNA composition and properties. As GC-rich regions are more resistant to decomposition and nuclease activity, their fraction accrues, especially rDNA (PMID: 27648955, 31467866). Interestingly, the GC-rich fraction has been shown to be especially bioactive (PMID: 31205871, 26273425, 27753029, 22594608). Another strongly activating modification is cfDNA oxidation (PMID: 32677958, 23644378, 27753029), that can be often found in pathology due to intrinsic oxidative stress. It suggests that oxidized/CG-enriched cfDNA can boost the inflammatory and other responses even in decreased or normal amounts. This is a serious limitation of the study.
Reply: Thank you these valuable suggestions. The studies you recommend have been helpful in discussing the findings. We have added information about the dynamic nature of cfDNA release and excretion to the manuscript (please see lines 277-282). In addition, we added data on the pro-inflammatory properties of CG-enriched cfDNA and oxidized cfDNA to section 4.2.
- Another limitation is considering multiple reports published by the same research team as independent reports. There are two studies by Kageyama et al., three by Ershova et al. and two reports by Lindqvist et al. The authors only found ‘asymmetry’ in three reports made by the same research group (Ershova et al.), but attributed it to using ‘a similar method’ (lines 205-207). The authors did not explain in the text what the ‘asymmetry’ means, but isn’t it obvious that using the same technique in not the only cause of the bias? Another factor can be recruitment of the same subjects, for example, in a series of studies, more recent studies performed on a large sample can include data obtained earlier on a smaller sample. So, some measurements in the series are repeated, not independent.
Reply: There are many reasons for funnel plot asymmetry (please see [Sterne J. A. C. et al. Recommendations for examining and interpreting funnel plot asymmetry in meta-analyses of randomised controlled trials. Bmj 343 (2011)]. We did not aim to explain the observed asymmetry. But in any case, the asymmetry indicates possible publication bias. We agree that some studies performed by the same research group may be dependent and have overlapping samples. Therefore, we have added information about a possible publication bias to the manuscript (please see lines 208-209 and 338-342).
- Finally, some minor suggestions to repair small defects:
5.1. I found some grammar mistakes. The text should be thoroughly edited by a native English speaker or a professional translator.
Reply: We made a few changes to the manuscript and checked the quality of the English language using the Digital Editing Tools. The Language Quality Score of our manuscript is “excellent”.
5.2. In lines 144-145, the authors declare that ‘The total number of individual studies included in the meta-analysis is sixteen because some reports considered several diseases simultaneously.’ But in Fig. 1 (PRISMA flowchart diagram), the last step box reads ‘Reports included in the meta-analysis (n=17)’. Was the report number 16 or 17? The discrepancy must be corrected.
Reply: The total number of individual studies included in the meta-analysis is sixteen because some reports considered several diseases simultaneously. In particular, the level of cf-mtDNA in BD and DDs was analyzed by Kageyama et al. Therefore, the total number of studies (8+4+5=17) differs from the number of individual studies included in the meta-analysis (16). In order not to mislead the reader, we corrected Figure 1 and removed the total number of studies. The total number of studies included in the meta-analysis is indicated in the text.
5.3. In ‘Depressive disorders’ part of Table 1, the study ‘Lindqvist et al. [45]’ is mentioned twice. Apparently, the study by Lindqvist et al. dated 2018 must be reference [47], not [45].
Reply: Indeed, it was a typo. We have corrected the reference number in the table.
Thanks again for your valuable suggestions.
Best regards
Authors